# New Antineoplastic Naphthohydroquinones Attached to Labdane and Rearranged Diterpene Skeletons

**DOI:** 10.3390/molecules26020474

**Published:** 2021-01-18

**Authors:** Ángela P. Hernández, Pablo Chamorro, Mª Lucena Rodríguez, José M. Miguel del Corral, Pablo A. García, Andrés Francesch, Arturo San Feliciano, Mª Ángeles Castro

**Affiliations:** 1Departamento de Ciencias Farmacéuticas, Área de Química Farmacéutica, Facultad de Farmacia, CIETUS/IBSAL, University of Salamanca, Campus Miguel de Unamuno, 37007 Salamanca, Spain; angytahg@usal.es (Á.P.H.); chamorrosanchez@gmail.com (P.C.); mlucenarodriguez@gmail.com (M.L.R.); jmmcs@usal.es (J.M.M.d.C.); pabloagg@usal.es (P.A.G.); artsf@usal.es (A.S.F.); 2Department of Medicine and General Cytometry Service-Nucleus, CIBERONC CB16/12/00400, Cancer Research Centre (IBMCC/CSIC/USAL/IBSAL), 37007 Salamanca, Spain; 3PharmaMar S.A., Avda de los Reyes, 1, 28770 Colmenar Viejo, Spain; afrancesch@pharmamar.com; 4Programa de Pós-Graduaçao em Ciências Farmacêuticas, Universidade do Vale do Itajaí, UNIVALI, 88302-202 Itajaí, SC, Brazil

**Keywords:** terpenylquinone, rearranged diterpenes, labdane, halimane, cytotoxicity

## Abstract

Terpenylquinones are mixed biogenesis primary or secondary metabolites widespread in Nature with many biological activities, including the antineoplastic cytotoxicity, that have inspired this work. Here, we present a cytotoxic structure-activity relationship of several diterpenylhydroquinone (DTHQ) derivatives, obtained from the natural labdane diterpenoid myrceocommunic acid used as starting material. Different structural modifications, that changed the functionality and stereochemistry of the decalin, have been implemented on the bicyclic core through epoxidation, ozonolysis or decarboxylation, and through induction of biomimetic breaks and rearrangements of the diterpene skeleton. All the isomers generated were completely characterized by spectroscopic procedures. The resulting compounds have been tested in vitro on cultured cancer cells, showing their relevant antineoplastic cytotoxicity, with GI_50_ values in the μM and sub-μM range. The rearranged compound **8** showed the best cytotoxic results, with GI_50_ at the submicromolar range, retaining the cytotoxicity level of the parent compounds. In this report, the versatility of the labdane skeleton for chemical transformation and the interest to continue using structural modifications to obtain new bioactive compounds are demonstrated.

## 1. Introduction

Terpenyl-quinones/hydroquinones (TQs/THQs) are mixed biogenesis primary or secondary metabolites widespread in Nature. Examples of primary metabolites are vitamin K, tocopherols, ubiquinones or plastoquinones that play important roles in the electron transport chain or in photosynthesis [1]. As secondary metabolites, TQs/THQs have been isolated from different sources, particularly of marine origin such as avarol or avarone (Figure 1), and they often have a wide variety of biological activities, mainly antineoplastic and antioxidant properties, that have been related to their involvement in redox cycling and/or Michael-1,4-addition reactions [2,3].

Our research group has been involved in the design, synthesis and biological evaluation of several series of TQs/THQs, derived either from the monoterpene β-myrcene or the diterpenoid myrceocommunic acid, giving rise to two series of compounds named monoterpenylquinones (MTQs/MTHQs) and diterpenylquinones (DTQs/DTHQs) (Figure 1). We have prepared a large number of derivatives with variations in sizes and functionalities of both the quinone and the terpenoid moieties. Very interesting antineoplastic compounds were obtained from β-myrcene [4,5,6,7,8,9,10] and myrceocommunic acid [11,12,13,14], being the DTQs/DTHQs slightly more potent than the MTQs/MTHQs. About the quinone size, 1,4-naphthoquinone (NQ) and 1,4-anthracenequinone (AQ) analogues were more potent than the corresponding benzoquinones (BQs) and 9,10-AQs, the common moiety in clinically used anticancer drugs, without significant differences between the quinone functions and the corresponding acetylated hydroquinones. Particularly interesting have been the results of several DTHQ-cyclolignan hybrids against the osteosarcoma cell line MG-63, which is a very aggressive type of cancer [14].

In addition to the cytotoxicity assays, several of our TQs/THQs were evaluated as antifungal [10,15], antiviral [10,16] and anti-leishmanial [17], attaining promising results against yeasts, filamentous fungi and *Leishmania infantum*.

In our previous research, many MTQs/MTHQs have been obtained by transformation of the side chain derived from the monoterpenoid [5,6,9,10], however only a few examples dealt with modifications of the diterpenoid core, always keeping the labdane type decalin [11,12]. Now, we describe further chemical transformations performed at this decalin moiety in the DTHQ series, either at the A or B decalin rings, including isomerizations, decarboxylations or rearrangements, leading to derivatives with modified diterpenoid skeletons, which have been evaluated against four tumor cell lines, representative of common and multi-drug resistant cancer types affecting lung, colon, breast and skin, to analyze the influence of those modifications on their cytotoxicity.

## 2. Results and Discussion

### 2.1. Chemistry

Continuing with our research on the field of terpenylquinones and considering the previous interesting results, we decided to enlarge the structural diversity of our DTHQ derivatives by chemical transformations on the decalin core, taking advantage of the functional groups present on it, i.e., the double bond around C-8 and the carboxylic group at C-4, making use of reactions such as epoxidation, ozonolysis and decarboxylation, which were applied to the previously reported DTHQs **1**–**4**.

The starting DTHQs **1** and **2** were prepared through a Diels-Alder cycloaddition between myrceocommunic acid or methyl myrceocommunate and *p*-benzoquinone (Scheme 1), following the procedure previously described by us [11,12]. Isomerization of the Δ^8(17)^ double bond, in both **1** and **2** to the more stable endocyclic and tetrasubstituted Δ^8^ double bond, was achieved by treatment with aq 57% HI at 80 °C to yield the DTHQs **3** and **4** [13].

#### 2.1.1. Epoxidation and Rearrangement Reactions

It is well known that the Lewis acid-catalyzed rearrangement of epoxides is an efficient tool to obtain new compounds through carbocation rearrangements [18]. In fact, in our previous work [11], the epoxidation of the Δ^8(17)^ double bond, followed by rearrangement promoted by BF_3_·Et_2_O, gave the corresponding aldehyde at C-17, with improved cytotoxicity respecting DTHQ **2**. Here in this work, in order to increase the structural diversity of our DTHQs, we explore the rearrangement products of the endocyclic Δ^8^ double bond epoxidation.

Treatment of **4** with MCPBA afforded a mixture of the two possible epoxides **5** and **6** in a 2:1 ratio. Both isomers were separated by column chromatography, characterized through their NMR spectra and treated separately with BF_3_·Et_2_O (Scheme 2).

Reaction of the β-epoxide **5** with the Lewis acid at −78 °C gave compounds **7**, **8** and the isomers **9a** and **9b**. The ^1^H- and ^13^C-NMR spectra of DTHQ **7** indicated the presence of the exocyclic double bond at C-8, similar to that present in DTHQs **1** and **2**, whereas the spectrum of **7** didn’t have the characteristic signal of the axial methyl group of a labdane type decalin. In fact, such signal moved from around 0.5 ppm to 1.19 ppm; this fact, together with the ^13^C-NMR data and the molecular formula C_31_H_36_O_6_ by HRMS, indicated us that the labdane decalin was rearranged to an halimane type decalin [19] with a tetrasubstituted Δ^5(10)^ double bond. The configuration at C-9 was deduced from the proposed epoxide rearrangement mechanism, probably via an unconcerted mechanism [19], that involved 1,2-shift of the axial methyl group at C-10 as illustrated by pathway “I” in Scheme 3.

The ^1^H and ^13^C-NMR spectra of DTHQ **8** showed signals for a ketone function and three methyl groups on aliphatic quaternary carbons. Several spectroscopic experiments (NOE, HMBC and HMQC and HRMS data) confirmed the contraction of the ring by migration of the chain following the pathway “II” shown in Scheme 3.

Compounds **9a** and **9b** showed to be regioisomers that practically coeluted in the CC, and only a small amount of **9a** was purified. They had a tertiary hydroxyl group and very similar spectroscopic properties, being the main difference in the ^1^H-NMR spectra, the presence or absence of an olefinic proton signal at 5.52 ppm. NOE, HMBC and HMQC experiments performed on **9a** allowed to locate the double bond at the Δ^1(10)^ position and to assign the *S* configuration for C-8; consequently, the double bond in **9b** should be placed in position Δ^5(10)^ and this compound would be the precursor of **7** through loss of a water molecule (Scheme 3).

Rearrangement of the α-epoxide **6** in the same conditions yielded a reaction product from which the ketone **10** and the alcohol **11** were isolated by CC (Scheme 2). The NMR data of **10** indicated the presence of a ketone (210.0 ppm) and two methyl groups on a double bond (1.62 and 1.69 ppm). HRMS, HMBC and HMQC experiments corroborated the opening of the B-ring of the decalin, and its degradation can be explained from the proposed rearrangement mechanism shown in Scheme 4. The *Z* configuration for the double bond was proposed on the basis of this mechanism and literature data for the ^13^C NMR chemical shift of methyl groups on double bonds [20]. 

Compound **11** had spectra data very similar to those of compound **9a**, with simple differences in the chemical shifts of the C-17 and C-20 methyl groups, thus indicating that **11** should be the C-8 epimer of **9a**.

#### 2.1.2. Ozonolysis

Ozonolysis is a well-known reaction for oxidative cleavage of carbon-carbon double bonds to get carbonyl compounds under reductive or oxidative conditions [21]. It has been widely used for structure elucidation of organic compounds, but also to get fragments with carbonyl groups. Interestingly, if the double bond is within a ring, a chain containing two carbonyls is obtained. 

In the work described here, DTHQs **1** and **2** have an exocyclic double bond and its ozonolysis under reductive work-up conditions gave respectively the ketones **12** and **13**, both having a 17-norlabdane decalin core (Scheme 5). However, DTHQ **4** has an endocyclic tetrasubstituted double bond and its ozonolysis afforded the 8,9-secolabdane diketone **14** and the epoxide **6** in a 1:1.5 ratio (Scheme 5). The formation of epoxides is frequent during ozonolysis of sterically hindered double bonds [22] as it happens in the case of **4**.

Diketone **14** was submitted to further intramolecular aldolic condensation in the presence of DBU giving rise to the rearranged labdane derivative **15** (Scheme 5). Under basic conditions, a partial saponification of the aromatic acetates was observed, so the reaction product was acetylated to restore the naphthohydroquinone diacetate moiety of the starting DTHQs. Both the secodecalin and the rearranged labdane derivative were characterized by spectroscopic comparison with the natural products chapecoderins A and B, which have the same labdane skeletons without the carboxylic function at C-19 [23].

#### 2.1.3. Decarboxylation Reactions

The last type of transformations performed on the DTHQs take advantage of the presence of the carboxylic group, which allows to get derivatives modified at the A-ring of the decalin. We obtained several 19-nor-DTHQs from **1** and **3** by decarboxylation with lead tetraacetate (LTA) [24]. Thus, the oxidative decarboxylation of **1** with LTA and cupric acetate in toluene-pyridine gave a 5:7:10 mixture (as deduced from the C-20 methyl signal the ^1^H-NMR spectrum) of the three unsaturated derivatives **16a**–**c** respectively, and the triacetate **17** [25] (Scheme 6). When this reaction product was treated with BF_3_·Et_2_O, the triacetate was transformed into the olefins **16a**–**c**. The three isomers **16a**–**c** were practically inseparable by CC, even using silica gel impregnated with silver nitrate, and only a small amount of the **16a**, which has two exocyclic methylenes, was separated. Consequently, the mixture **16a**–**c** was treated with HI to attempt the isomerization to the more stable Δ^4^ and Δ^8^ endocyclic double bonds. However, the isomerization was not complete and a mixture of **18b** and **18c** was obtained (Scheme 6). To force the isomerization towards **18c**, the mixture **18b**,**c** was treated with iodine in toluene for longer time, however, a complex reaction product was obtained from which the rearranged aromatic derivative **19** was separated and characterized (Scheme 6). The isomer **18c** was also obtained after treatment of DTHQ **3** with LTA followed by reaction with iodine for 3 h (Scheme 6).

The oxidative decarboxylation was also applied to DTHQ **12** to afford the ketoacetate **20** and the isomers **21a**–**c** (Scheme 3). When the mixture **21a**–**c** was treated with iodine, the isomer **21c**, having the most stable tetrasubstituted double bond, was isolated and characterized.

The structure of compound **19** was deduced from the analysis of its spectroscopic properties. Among the most characteristic signals in its ^1^H-NMR spectrum, those corresponding to three methyl groups stood out, one doublet at 1.30 ppm and two singlets at 2.35 and 2.23 ppm, the latter assignable to methyls attached to an aromatic ring. Additional spectroscopic experiments (^13^C-NMR, HRMS, NOE, HMBC and HMQC) confirmed the structure of **19**, whose formation can be explained by the proposed mechanism shown in Scheme 7, in which the observed NOEs for **19** are shown with blue arrows, although the absolute configuration at C-4 could not be determined. Similar rearrangements towards B-ring aromatization have also been described for other natural labdanes [26].

### 2.2. Biological Evaluation

Cell killing ability of the synthetized DTHQ derivatives was conducted against a panel of three human tumor cell lines with high incidence in the population [27], including non-small cell lung carcinoma (A-549), colon adenocarcinoma (HT-29), and multi-drug-resistant breast carcinoma (MDA-MB-231). The in vitro antiproliferative activity was assessed using the colorimetric SRB (sulforhodamine B) method [28] and the common anticancer naphthacenequinone drug doxorubicin was included in the assays as reference. The GI_50_ (μM) values (concentration that causes 50% growth inhibition), for all the tested compounds are shown in Table 1. Additionally, compounds **1**, **2**, **8**, and **21a** were also assayed on malignant skin melanoma SK-MEL-28 cells with GI_50_ values of 0.24, 0.12, 0.54 and 1.66 μM respectively.

From these results, it can be stated that the compounds assayed were fairly cytotoxic for neoplastic cells. Indeed, some of them showed GI_50_ values similar to those of doxorubicin. In general, the presence of oxygenated functions, the opening, contraction or rearrangement of the decalin moiety gave compounds that kept or slightly decreased the potency of the lead compounds **1** and **2**.

There were no differences in potency between compounds with exocyclic or endocyclic double bond in the B-ring of the decalin core (compounds **1**–**4**). The presence of a ketone in such ring led to less potent compounds (**12**, **13** and **21** vs. **1** and **2**), while DTHQ **12**, with a free carboxylic group, was the less potent compound of the series on the three cell lines. Interestingly, the α-epoxide **6** was four times more potent than the β-epoxide **5**, what is in accordance with previous results obtained with the Δ^8(17)^ epoxides [11]. Those analogues with a halimane type decalin and a hydroxyl group at C-8 (**9** and **11**), were less potent than their precursors, though keeping their cytotoxicity values in the micromolar range. The same happened with DTHQs **10** and **14**, which lacked the decalin B-ring. Several differences were observed for DTHQs **8** and **15**, which had an indane system, being **8** as potent as its precursors, while **15** was fairly less potent. Such a remarkable difference, of more than one order of magnitude, would be accounted for by an intramolecular hydrogen bonding between the carbonyl and the hydroxyl group attached to the cyclopentane ring in **15**, which would prevent potential interactions with hypothetical targets, unlike the free ketone in **8**. The presence of the ketone out of the bicyclic core in **8** may also have a positive impact on activity, since other decalin compounds with ketone groups located on the ring, such as **12**, **13** or **21**, resulted less potent than the indanyl ketone **8**.

The decarboxylated analogues were, in general, less potent than the parent compounds; however, DTHQ **18** had the same GI_50_ value than its precursor without differences between the **a**–**c** isomers, and DTHQs **17** and **20**, with an acetoxy group at C-4, retained the submicromolar cytotoxicity.

Finally, in terms of cytotoxic potency, these results indicated that a varied range of functionalization in the B-ring was tolerated and that the ester function in the A-ring seemed to be important for the antineoplastic cytotoxicity.

## 3. Materials and Methods

### 3.1. Chemistry

NMR experiments were recorded on a Bruker WP 200 SY (Bruker GmbH, Karlsruhe, Germany) (200 and 50.3 MHz for ^1^H and ^13^C) or Bruker Avance 400DRX (Bruker GmbH, Karlsruhe, Germany) (400 and 100 MHz) spectrometers in CDCl_3_ using TMS as internal reference. Chemical shift (δ) values are expressed in ppm followed by multiplicity and coupling constants (*J*) in Hz. Optical rotations were recorded on a Perkin-Elmer 241 polarimeter (Perkin-Elmer GmbH, Uberlingen, Germany) in CHCl_3_ solution. UV spectra were obtained on a Hitachi 100–60 spectrophotometer (Hitachi High-Technologies Corporation, Tokyo, Japan) in ethanol solution, and λ_max_ are given in nm. IR spectra were obtained on a Nicolet Impact 410 spectrophotometer (Nicolet Instrument Corporation, Madison, WI, USA) in NaCl film. HRMS were run in a VG TS-250 spectrometer (VG Instruments, Manchester, UK) working at 70 eV, using electrospray ionization (ESI) or fast atom bombardment (FAB). Elemental analysis (C, H, N) were obtained with a PERKIN-ELMER 2400 CHN (Perkin-Elmer GmbH, Uberlingen, Germany). Solvents and reagents were purified by standard procedures as necessary. Column chromatography (CC) purifications were performed using silica gel 60 (40–63 mm, 230–400 mesh, Merck) and TLC was carried out on silica gel 60 F_245_ (Merck, 0.25 mm thick). NMR assignments are included in the Appendix A.

Diterpenylnaphthohydroquinone diacetates **1**, **2** and **4** were obtained by means of previously described procedures [11,12]. Other compounds were prepared as follows.

DTHQ **3**. To a 10^−2^ M solution of **1** (185 mg, 0.38 mmol) in toluene (37 mL) aq HI 57% (0.86 mL) was added. The mixture was stirred at 80 °C for 10 min. Then, ethyl acetate was added and the organic layer was washed with aq satd NaHCO_3_ and brine, dried over Na_2_SO_4_, filtered and evaporated off yielding the Δ^8^ product **3** (182 mg, 0.37 mmol, 98%): IR cm^−1^ (film): 2930, 1765, 1690, 1450, 1365, 1200, 1180, 1050, 820, 735. Anal. Calcd. for C_30_H_36_O_6_: C,73.15; H, 7.37. Found C, 73.12%; H, 7.36%. RMN ^1^H: Table 2. RMN ^13^C: Table 3.

DTHQ epoxides **5** and **6**. Compound **4** (178 mg, 0.35 mmol) was dissolved in dichloromethane and MCPBA (485 mg, 1.55 mmol) and NaHCO_3_ (528 mg) were added. The mixture was kept at room temperature for 3 h. Then dichloromethane was added, followed by 10% aqueous Na_2_S_2_O_3_ until the oxidant was eliminated. The organic layer was washed with brine and dried over anhydrous Na_2_SO_4_, and the solvent evaporated off. The reaction product was purified by CC (CHCl_3_/EtOAc 9:1) to obtain: 65 mg (35%) of epoxide **5**, 42 mg (23%) of mixture of **5** and **6** and 28 mg (15%) of epoxide **6**. Compound **5**: [α${]}_{D}^{22}$ = +50.9 (c, 0.97%); IR cm^−1^ (film): 2945, 1764, 1717, 1462, 1433, 1366, 1262, 1203, 1182, 1053, 891, 825; UV λ_max_ (ε): 240 (12300), 287 (5500). HRMS-FAB: calcd for C_31_H_39_O_7_ [M + H]^+^ 523.2696; found 523.2742 *m*/*z*. RMN ^1^H: Table 2. RMN ^13^C: Table 3. Compound **6**: [α${]}_{D}^{22}$ = +25.1 (c, 1.0%); IR cm^−1^ (film):2920, 2850, 1765, 1720, 1610, 1465, 1365, 1200, 1180, 1050, 890, 825, 735; HRMS-FAB: calcd for C_31_H_39_O_7_ [M + H]^+^ 523.2696 u; found 523.2698 *m*/*z*; RMN ^1^H: Table 2. RMN ^13^C: Table 3.

Rearranged DTHQs **7**, **8** and **9.** To a solution of epoxide **5** (62 mg, 0.12 mmol) in dry dichloromethane, cooled at −78 °C, BF_3_-Et_2_O (55 μL, 0.43 mmol) was added under inert atmosphere. The mixture was stirred at this temperature for 3 h. Then the solvent was removed, and the crude dissolved in EtOAc, washed with brine, dried over Na_2_SO_4_, and the solvent evaporated off. The reaction crude was chromatographed to yield: 6 mg (9%, eluent: Hex/EtOAc, 9:1) of **7**, 24 mg (38%, eluent: Hex/EtOAc, 9:1) of **8**, 7 mg (11%, eluent: Hex/EtOAc, 7:3) of **9a** and 17 mg (27%, eluent: Hex/EtOAc, 7:3) of **9a**,**b**. Compound **7**: IR cm^−1^ (film): 2935, 1765, 1725, 1610, 1450, 1365, 1200, 1050, 1010, 895, 825, 735; HRMS-FAB: calcd for C_31_H_37_O_6_ [M + H]^+^ 505.2590; found 505.2623 *m*/*z*. RMN ^1^H: Table 2. RMN ^13^C: Table 3. Compound **8**: [α${]}_{D}^{22}$ = +19.7 (c, 0.97%); IR cm^−1^ (film): 2495, 1765, 1720, 1625, 1610, 1450, 1365, 1200, 1050, 1005, 895, 825, 735; HRMS-FAB: calcd for C_31_H_39_O_7_ [M+H]^+^ 523.2696; found 523.2700 *m*/*z*. RMN ^1^H: Table 2. RMN ^13^C: Table 3. HMQC and HMBC experiments: Appendix A. Compound **9a**: IR cm^−1^ (film): 3515, 2935, 1765, 1610, 1450, 1365, 1265, 1200, 1120, 1010, 915, 820, 730; HRMS-FAB: calcd for C_31_H_37_O_6_ [M+H]^+^ 505.2590; found 505.2623 *m*/*z*. RMN ^1^H: Table 2. RMN ^13^C: Table 3. HMQC and HMBC experiments: Appendix A.

Rearranged DTHQs **10** and **11**. To a solution of epoxide **6** (89 mg, 0.17 mmol) in dry dichloromethane, cooled at −78 °C, BF_3_-Et_2_O (78 μL, 0.61 mmol) was added under inert atmosphere. The mixture was stirred at this temperature for 3 h. Then the solvent was removed, and the crude dissolved in EtOAc, washed with brine, dried over Na_2_SO_4_, and the solvent evaporated off. The reaction crude was chromatographed to yield: 25 mg (28%, eluent: Hex/EtOAc, 7:3) of **10** and 29 mg (32%, eluent: Hex/EtOAc, 6:4) of **11**. Compound **10**: [α${]}_{D}^{22}$ = +2.8 (c, 0.65%); IR cm^−1^ (film): 2948, 1765, 1727, 1608, 1462, 1433, 1366, 1020, 1182, 1053, 897, 825, 735. HRMS-FAB: calcd for C_31_H_39_O_7_ [M + H]^+^: 523.2696; found 523.2674 *m*/*z*. Anal. Calcd. for C_31_H_38_O_7_: C,71.24; H, 7.33. Found C, 71.19%; H, 6.90%. RMN ^1^H: Table 2. RMN ^13^C: Table 3. HMQC and HMBC experiments: Appendix A. Compound **11**: [α${]}_{D}^{22}$ = −62.4 (c, 0.65%); IR cm^−1^ (film): 3536, 29,36, 1765, 1728, 1609, 1462, 1433, 1365, 1203, 1182, 1054, 1016, 939, 899, 736. UV λ_max_ (ε): 240 (12600), 286 (4600). HRMS-ESI: calcd for C_31_H_38_O_7_Na [M + Na]^+^: 545.2509; found 545.2528 *m*/*z*. RMN ^1^H: Table 2. RMN ^13^C: Table 3. HMQC and HMBC experiment: Appendix A. 

General procedure for ozonolysis. nor-DTHQ **12**. A solution of compound 1 (340 g, 0.69 mmol) in dry CH_2_Cl_2_ (6 mL) was cooled to –78 °C, and ozone was bubbled (30 nL/h, 0.6 amperes) through the mixture for 10 min. Then the reaction mixture was purged with oxygen for 10 min and argon for additional 5 min to remove the ozone excess. Then, dimethyl sulfide (1.5 mL, 19 mmol) was added to the mixture and stirred for 6 h. The solvent was distilled off, the residue was dissolved in EtOAc and washed with brine. The organic layer was dried over anhydrous Na_2_SO_4_ and evaporated off to give a reaction product that was purified by CC on silica-gel (eluent: Hex/EtOAc, 7:3) to yield **12** (227 mg, 66%). IR cm^−1^ (film): 2950, 1765, 1705, 1695, 1610, 1450, 1365, 1200, 1050, 1010, 915, 895, 735. RMN ^1^H: Table 2. RMN ^13^C: Table 3. HRMS-FAB: calcd for C_29_H_35_O_7_ [M + H]^+^: 495.2383; found 495.2377. 

nor-DTHQ **13**. Starting from **2** (220 mg, 0.44 mmol) and following the above procedure for ozonolysis, the reaction product yielded **13** (212 mg, 96%). IR cm^−1^ (film): 2935, 1765, 1720, 1710, 1610, 1450, 1365, 1200, 1050, 825, 735. HRMS-FAB: calcd for C_30_H_37_O_7_ [M + H]^+^: 509.2539; found 509.2602 *m*/*z*. RMN ^1^H: Table 2. RMN ^13^C: Table 3. 

seco-DTHQ **14**. Starting from **4** (171 mg, 0.34 mmol) and following the above procedure for ozonolysis, the reaction product was chromatographed to yield **6** (74 mg, 40%, eluent: Hex/EtOAc, 7:3) and **14** (49 mg, 27%, eluent: Hex/EtOAc, 8:3). [α${]}_{D}^{22}$ = +19.0 (c, 0.97%); IR cm^−1^ (film): 2945, 1764, 1717, 1462, 1433, 1366, 1262, 1203, 1182, 1053, 891, 825. UV λ_max_ (ε): 240 (12300), 289 (4500). HRMS-FAB: calcd for C_31_H_39_O_8_ [M + H]^+^: 539.2645; found 539.2618 *m*/*z*. RMN ^1^H: Table 4. RMN ^13^C: Table 5. HMQC and HMBC experiments: Appendix A.

Rearranged DTHQ **15**. To a stirred solution of **14** (59 mg, 0.11 mmol) in dry toluene (3 mL) was added 1,8-diazabicyclo[5.4.0]undec-7-ene (DBU) (27 μL, 0.16 mmol). The reaction mixture was stirred for 6 h at 60 °C under nitrogen atmosphere. The reaction was quenched by addition of 2 M HCl and extracted with ethyl acetate. The combined organic layer was washed with brine and dried over anhydrous Na_2_SO_4_. The reaction product was acetylated with acetic anhydride in pyridine and then purified by column chromatography (eluent: Hex/EtOAc, 7:3) to yield unreacted **14** (45%) and **15** (30 mg, 51%). [α${]}_{D}^{22}$ = +8.6 (c, 1.0%); IR cm^−1^ (film): 3421, 2932, 1764, 1721, 1687, 1609, 1461, 1433, 1365, 1020, 1181, 1053, 896, 825. UV λ_max_ (ε): 240 (12300), 285 (5900). HRMS-FAB: calcd for C_31_H_39_O_8_ [M + H]^+^: 539.2645; found 539.2642 *m*/*z*. RMN ^1^H: Table 4. RMN ^13^C: Table 5. HMQC and HMBC experiment: Appendix A.

General procedure for decarboxylation with lead tetracetate. nor-DTHQs **16** and **17**. A solution of **1** (256 mg, 0.52 mmol) and pyridine (1.3 mL, 16 mmol) in dry toluene (10 mL) was heated at 145 °C for 1 h. Then lead tetraacetate (299 mg, 0.67 mmol) and copper diacetate (104 mg, 0.52 mmol) were added and kept stirred at the same temperature for additional 4.5 h. The cooled mixture was evaporated and redissolved in EtOAc. The organic layer was washed consecutively with 2 M HCl and brine and dried over anhydrous Na_2_SO_4_. The reaction product was purified by column chromatography to yield **16a**–**c** (109 mg, 47%, eluent: Hex/EtOAc, 95:5) and **17** (53 mg, 20%, eluent: Hex/EtOAc, 9:1). RMN ^1^H: Appendix A. RMN ^13^C: Appendix A. The mixture **16a**–**c** was chromatographed over silica gel impregnated with 20% silver nitrate, using Hex/EtOAc, 7:3 as eluent, to yield **16a**–**c** (63 mg, 47%, eluent:) and **16a** (21 mg. Compound **16a**: [α${]}_{D}^{22}$ = +10.5 (c, 0.54%); IR cm^−1^ (film): 3400, 2929, 2849, 1766, 1737, 1720, 1643, 1461, 1440, 1366, 1020, 1181, 1053, 890. UV λ_max_ (ε): 240 (5700), 284 (2100). HRMS-FAB: calcd for C_29_H_35_O_4_ [M + H]^+^: 447.2535; found 447.2516 *m*/*z*. RMN ^1^H: Table 4. RMN ^13^C: Table 5. Compound **17**: [α${]}_{D}^{22}$ = +11.3 (578), (c, 0.99%) IR cm^−1^ (film): 3078, 2935, 2870, 1766, 1724, 1641, 1608, 1432, 1366, 1252, 1020, 1181, 1053, 1013, 893, 826, 736. UV λ_max_ (ε): 240 (11400), 285 (4800). RMN ^1^H: Table 4. RMN ^13^C: Table 5.

nor-DTHQ **18**. Starting from **3** (250 mg, 0.51 mmol) and following the above procedure for decarboxylation, the CC of the reaction product over AgNO_3_ impregned silicagel (eluent: Hex/EtOAc, 9:1), yielded **18a** (23 mg, 10%), a 1:1:1 mixture of **18a**–**c** (64 mg, 28%) and **18c** (25 mg, 11%). Compound **18a**: [α${]}_{D}^{22}$ = +62.1 (c, 0.40%); IR cm^−1^ (film): 2930, 1765, 1610, 1450, 1365, 1200, 1180, 1050, 1010, 890, 820, 735. HRMS-FAB: calcd for C_29_H_35_O_4_ [M + H]^+^: 447.2535; found 447.2563 *m*/*z*. RMN ^1^H: Table 4. RMN ^13^C: Table 5. Compound **18c**: [α${]}_{D}^{22}$ = +99.7 (c, 0.76%); IR cm^−1^ (film): 2930, 1765, 1610, 1450, 1365, 1200, 1180, 1050, 1010, 895, 825, 735. HRMS-FAB: calcd for C_29_H_35_O_4_ [M + H]^+^: 447.2535; found 447.2542 *m*/*z*. RMN ^1^H: Table 4. RMN ^13^C: Table 5.

The mixture **18a**–**c** was dissolved in toluene and iodine (10 mg) was added and stirred at 95 °C for 3 h. The cooled mixture was evaporated and redissolved in EtOAc. The organic layer was washed consecutively with aq sat sodium thiosulphate (Na_2_S_2_O_3_) and brine and dried over anhydrous Na_2_SO_4_, to yield **18c** (96%).

Treatment of the mixture **16a**–**c** with aq HI 57% (following the procedure described above to obtain **3**), yield a 1:1 mixture of **18b** and **18c** (66%) after CC of the reaction product, eluted with Hex/EtOAc 95:5.

nor-DTHQ **19**. The mixture **18b**–**c** (116 mg, 0.26 mmol) was dissolved in toluene (7 mL) and iodine (44 mg, 0.17 mmol) was added and stirred at 95 °C for 24 h. The cooled mixture was evaporated and redissolved in EtOAc. The organic layer was washed consecutively with aq sat Na_2_S_2_O_3_ and brine and dried over anhydrous Na_2_SO_4_, to yield a reaction crude that was purified by CC to yield **19** (37 mg, 32%, eluent: Hex/EtOAc, 95:5). [α${]}_{D}^{22}$ = −1.7 (c, 0.9%); IR cm^−1^ (film): 2926, 2855, 1766, 1608, 1461, 1431, 1364, 1201, 1180, 1054, 1007, 894, 823. UV λ_max_ (ε): 241 (15100), 283 (4900). MS (*m*/*z*): 444 (M)^+^(10). RMN ^1^H: Table 4. RMN ^13^C: Table 5. HMQC experiment: Appendix A.

dinor-DTHQs **20** and **21.** Starting from **12** (230 mg, 0.46 mmol) and following the above procedure for decarboxylation, the CC of the reaction product yielded **21a** (7 mg, 3%, eluent: Hex/EtOAc, 95:5), **21a**–**c** (85 mg, 41%, eluent: Hex/EtOAc, 95:5) and **20** (15 mg, 7%, eluent: Hex/EtOAc, 9:1). Compound **20**: [α${]}_{D}^{22}$ = −16.5 (c, 0.96%); IR cm^−1^ (film): 2940, 1765, 1725, 1710, 1365, 1250, 1200, 1180, 1050, 1010. HRMS-ESI: calcd for C_30_H_36_O_7_Na [M + Na]^+^: 531.2353; found 531.2346 *m*/*z*. RMN ^1^H: Table 4. RMN ^13^C: Table 5. Compound **21a**: [α${]}_{D}^{22}$ = −1.6 (c, 0.67%); IR cm^−1^ (film): 2936, 2865, 1765, 1707, 1643, 1608, 1433, 1366, 1202, 1182, 1053, 1009, 895, 827, 735. UV λ_max_ (ε): 240 (8800), 285 (4000). HRMS-FAB: calcd for C_28_H_33_O_5_ [M + H]^+^: 449.2328; found 449.2330 *m*/*z*. RMN ^1^H: Table 4. RMN ^13^C: Table 5.

The mixture **21a**–**c** (85 mg, 0.17 mmol) was dissolved in toluene and iodine (20 mg) was added and stirred at 95 °C for 3 h. The cooled mixture was evaporated and redissolved in EtOAc. The organic layer was washed consecutively with aq sat Na_2_S_2_O_3_ and brine, and dried over anhydrous Na_2_SO_4_. CC of the reaction product yielded **21c** (27 mg, 32%, eluent: Hex/EtOAc, 9:1). Compound **21c**: IR cm^−1^ (film): 2935, 1765, 1705, 1600, 1430, 1365, 1200, 1180, 1050, 1005, 900, 825. HRMS-FAB: calcd for C_28_H_33_O_5_ [M + H]^+^: 449.2328 u; found 449.2342 *m*/*z*. RMN ^1^H: Table 4. RMN ^13^C: Table 5.

### 3.2. Biological Evaluation

A colorimetric type of assay using sulforhodamine B (SRB) reaction has been adapted for a quantitative measurement of cell growth and viability, following a previously described method [28]. This assay employs 96 well cell culture microplates of 9 mm diameter. Cell lines are obtained from American Type Culture Collection (ATCC) derived from different human cancer types. Cells are maintained in RPMI 1640 10% Fetal Bovine Serum (FBS), supplemented with 0.1 g/L penicillin and 0.1 g/L streptomycin sulfate and then incubated at 37 °C, 5% CO_2_ and 98% humidity. For the experiments, cells were harvested from subconfluent cultures using trypsin and resuspended in fresh medium before plating.

Cells were seeded in 96 well microtiter plates, at 5 × 10^3^ cells per well in aliquots of 195 μL of RPMI medium, and they were allowed to attach to the plate surface by growing in drug free medium for 18 h. Afterward, samples were added in aliquots of 5 μL in a ranging from 10 to 10^−8^ μg/mL, dissolved in DMSO:EtOH:Phosphate Buffered Saline (PBS) (0.5:0.5:99). After 72 h exposure, the antitumor effect was measured by the SRB methodology: cells were fixed by adding 50 μL of cold 50% (wt/vol) trichloroacetic acid (TCA) and incubating for 60 min at 4 °C. Plates were washed with deionised water and dried; 100 μL of SRB solution (0.4% wt/vol in 1% acetic acid) was added to each microtiter well and incubated for 10 min at room temperature. Unbound SRB was removed by washing with 1% acetic acid. Plates are air-dried and bound stain was solubilized with tris(hydroxymethyl)aminomethane (Tris) buffer. Optical densities (OD) were read on an automated spectrophotometric plate reader at a single wavelength of 490 nm. Data analyses were generated automatically by LIMS implementation. Using control OD values (C), test OD values (T) and time zero OD values (T_0_), the drug concentration that causes 50% Growth Inhibition (GI_50_ value) was calculated from the equation: 100 × [(T − T_0_)/(C − T_0_)] = 50. Each value represents the mean from triplicate determinations.

## 4. Conclusions

In this work, we reported several modifications of the decalin core of DTHQs obtained from the natural labdanic diterpenoid myrceocommunic acid. With the aim to enlarge the structural diversity and the number of DTHQ derivatives, we have carried out certain chemical transformations into the natural bicyclic diterpene core, such as epoxidation, ozonolysis or decarboxylation, taking advantage of the functional groups present on it, that are, the double bond around C-8 and the carboxylic group at C-4. Considering this, we have obtained a collection of novel compounds that included some derivatives rearranged towards different diterpenic skeletons as halimane, norlabdane or secoditerpene analogues.

These chemical approaches illustrate the versatility of the labdane ring and show the complex reactivity of the system. Despite the chemical complexity of some rearrangements, most of the final molecules obtained, were thoroughly characterized and their cytotoxicity tested on common solid tumor cell lines. According to the GI_50_ values obtained from the in vitro evaluation, all the compounds tested maintained their cytotoxicity in the µM level, showing no changes in potency when the doble bound in the decalin B-ring was isomerized, but the conservation of the terpenic bicyclic system, either decalin or indane, in these molecules seems to be important. Additionally, oxygenated functions such as the ester group in the A-ring appears to be a main point for the activity of this family of compounds, without forgetting that its cytotoxicity would also be associated with the hydroquinonic part of the molecule, which is identical in all the compounds. These results lay the foundations for the introduction of further functional changes in the decalin moiety that would make possible to obtain more potent and effective antineoplastic compounds.

## Data Availability

The data presented in this study are available in this article.

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
