# Peer review of "New Antineoplastic Naphthohydroquinones Attached to Labdane and Rearranged Diterpene Skeletons"

_molecules, 2021, doi:10.3390/molecules26020474_

Round 1

Reviewer 1 Report

New antineoplastic naphthohydroquinones attached to labdane and rearranged diterpene skeletons

Abstrac:  

It does not mention which is the one with the lowest GI50, in the synthesis it does not mention the spectroscopic difficulty of the isomers generated.

Introduction:

It remains to explain why they used these cell lines.

lung carcinoma (A-549), colon carcinoma (HT-29), malign skin melanoma (SK-MEL-28) and multi-drug resistant breast carcinoma (MDA-MB-231).

Results and discussion

Biological evaluation

Table 1, SK-MEL-28 and MDA-MB-231 results are missing.

Explain why compound 8 has excellent head and hit characteristics.

Carry out SAR with the data obtained.

Materials and Methods

All compounds must be described in 1H and 13C NMR. For example.

1H NMR (CDCl3, 400 MHz) δ 8.11 (1H, d, J = 7.6 Hz, H-11), 7.54–7.28 (2H, m, H-9, H-10), 7.27 (1H, m, H-8), 6.59 (1H, s, H-3), 6.10, 5.97 (each 1H, d, J = 1.5 Hz, OCH2O), 4.86 (1H, dd, J = 13.7, 4.4 Hz, H-6a), 4.44 (1H, m, H-5a), 3.77 (3H, s, NCOOCH3), 3.06 (1H, m, H-7a), 2.99 (1H, m, H-5b), 2.91 (1H, m, H-7b), 2.82 (1H, m, H-4a), 2.61 (1H, m, H-4b); 13C NMR (CDCl3, 100 MHz) δ 155.8 (C, NCOOCH3), 146.8 (C, C-2), 143.0 (C, C-1), 135.8 (C, C-7a), 130.7 (C, C-11a), 128.7 (CH, C-8), 127.79 (C, C-3a), 127.78 (CH, C-9), 127.2 (CH, C-10), 127.0 (CH, C-11), 125.6 (C, C-3b), 117.3 (C, C-1a), 107.6 (CH, C-3), 100.9 (CH2, OCH2O), 52.7 (CH3, NCOOCH3), 51.7 (CH, C-6a), 39.2 (CH2, C-5), 34.5 (CH2, C-7), 30.4 (CH2, C-4). 

Supplementary Materials

Include all NMR spectra. (Figure asignated).

Author Response

Answer to Reviewer 1

- Abstract:

It does not mention which is the one with the lowest GI50, in the synthesis it does not mention the spectroscopic difficulty of the isomers generated.

According to this suggestion, two sentences mentioning the most potent compound and the complete characterization of the different isomers have been included in the Abstract.

- Introduction:

It remains to explain why they used these cell lines. lung carcinoma (A-549), colon carcinoma (HT-29), malign skin melanoma (SK-MEL-28) and multi-drug resistant breast carcinoma (MDA-MB-231).

                The reason to choose these cell lines is because the tumours they represent (lung, colon, melanoma and breast) have a high incidence in the population. This reason has been mentioned in the last paragraph of the Introduction (line 71) and at the beginning of the Results and Discussion section (line 207) with a reference (nº 27) to the USA National Cancer Institute.

- Results and discussion

Biological evaluation

Table 1, SK-MEL-28 and MDA-MB-231 results are missing.

Explain why compound 8 has excellent head and hit characteristics.

Carry out SAR with the data obtained.

                Considering the number of compounds tested on the two cell lines omitted in Table 1, the results for MDA-MB-231 have been included therein. However, results for SK-MEL-28 cell line were obtained for only four compounds, so we just decided to include the GI50 values in the text so as not to see many empty spaces in Table 1.

                Related to the indane derivative 8, actually the most potent compound of the series, there isn’t any clear explanation for its properties; differences with the other indane derivative 15 could be explained through possible hydrogen bonding and in this sense, several SAR dealing statements have been added in lines 231-238 in the following terms:

Such a remarkable difference, of more than one order of magnitude, would be accounted for by an intramolecular hydrogen bonding between the carbonyl and the hydroxyl group attached to the cyclopentane ring in 15, which would prevent potential interactions with hypothetical targets, unlike the free ketone in 8. The presence of the ketone out of the bicyclic core in 8 may also have a positive impact on activity, since other decalin compounds with ketone groups located on the ring, such as 12, 13 or 21, resulted less potent than the indanyl ketone 8

                As just said, some SAR analyses were already done in the discussion and also mentioned in the conclusions section (lines 558, 561). In our opinion no more reliable SAR comparisons can be made or new conclusions added, the compounds are structurally similar and it should also be taken into account that the cytotoxicity of this family of compounds would be also associated with the hydroquinonic part of the molecule, which is identical in all the compounds.

- Materials and Methods

 All compounds must be described in 1H and 13C NMR. For example.

1H NMR (CDCl3, 400 MHz) δ 8.11 (1H, d, J = 7.6 Hz, H-11), 7.54–7.28 (2H, m, H-9, H-10), 7.27 (1H, m, H-8), 6.59 (1H, s, H-3), 6.10, 5.97 (each 1H, d, J = 1.5 Hz, OCH2O), 4.86 (1H, dd, J = 13.7, 4.4 Hz, H-6a), 4.44 (1H, m, H-5a), 3.77 (3H, s, NCOOCH3), 3.06 (1H, m, H-7a), 2.99 (1H, m, H-5b), 2.91 (1H, m, H-7b), 2.82 (1H, m, H-4a), 2.61 (1H, m, H-4b); 13C NMR (CDCl3, 100 MHz) δ 155.8 (C, NCOOCH3), 146.8 (C, C-2), 143.0 (C, C-1), 135.8 (C, C-7a), 130.7 (C, C-11a), 128.7 (CH, C-8), 127.79 (C, C-3a), 127.78 (CH, C-9), 127.2 (CH, C-10), 127.0 (CH, C-11), 125.6 (C, C-3b), 117.3 (C, C-1a), 107.6 (CH, C-3), 100.9 (CH2, OCH2O), 52.7 (CH3, NCOOCH3), 51.7 (CH, C-6a), 39.2 (CH2, C-5), 34.5 (CH2, C-7), 30.4 (CH2, C-4).

                The 1H and 13C NMR data, placed in the supplementary material in the first version, have now been included in Tables 2-5 in the experimental section. We believe that tables result more visual and better for comparing the chemical shifts of similar compounds. Tables for HMBC and HMQC experiments remain in the supplementary material.

- Supplementary Materials

Include all NMR spectra. (Figure asignated).

All the NMR spectra have been included in the supplementary material as suggested by the Reviewer

Reviewer 2 Report

Hernández et al., describe the antineoplastic cytotoxicity of several modification compounds obtained from the natural labdanic diterpenoid myrceocommunic acid.
The article describes in detail the chemical modifications to which the investigated substances were subjected. Some conclusions are also made regarding the structure-activity relationship. The Supplementary Materials file contains tables with NMR data for the synthesized compounds.
The manuscript is well-written and structures and reactions are thoroughly elucidated. This publication will be of interest to both chemists and bio-researchers.
Overall, I recommend the acceptance of the manuscript after a minor revision .
The introduction section needs a Figure showing the main structural types (TQs/THQs, MTQs/MTHQs, DTQs/DTHQs)
Lines 281, 329, 379: Correct to °C.

Author Response

-Hernández et al., describe the antineoplastic cytotoxicity of several modification compounds obtained from the natural labdanic diterpenoid myrceocommunic acid.
The article describes in detail the chemical modifications to which the investigated substances were subjected. Some conclusions are also made regarding the structure-activity relationship. The Supplementary Materials file contains tables with NMR data for the synthesized compounds.
The manuscript is well-written and structures and reactions are thoroughly elucidated. This publication will be of interest to both chemists and bio-researchers.
Overall, I recommend the acceptance of the manuscript after a minor revision .

We appreciate these kind comments very much.

The introduction section needs a Figure showing the main structural types (TQs/THQs, MTQs/MTHQs, DTQs/DTHQs)

As suggested, a figure in the Introduction has been included with examples of the main structural types

Lines 281, 329, 379: Correct to °C.

They were corrected.

Reviewer 3 Report

This manuscript written by Hernández et al. shows the structure-cytotoxic activity relationship of several diterpenylhydroquinone derivatives using four cancer cell lines. A couple of compounds synthesized in this study maintain the cytotoxic activity at the similar level to that of the lead compound. This reviewer thinks that the chemistry part looks good, but biology part needs more information.

The authors used four cell lines; however, they show the cytotoxic data obtained from two cell lines only in Table 1. This reviewer recommends the authors to summarize all cytotoxic data obtained from four cell lines in Table 1.

The authors cultured cells in RPMI 1640 medium, which is not the optimal medium for the cell lines they used. So, this reviewer thinks that the data were obtained under a less common condition. By the way, did the authors culture cells without serum?

The authors should refer to the cytotoxic activity of the compounds on normal (non-cancer) cells because the title contains "antineoplastic." It is of importance to show that the compounds kill cancer cells specifically or nonspecifically.

Author Response

Answer to Review 3

-This manuscript written by Hernández et al. shows the structure-cytotoxic activity relationship of several diterpenylhydroquinone derivatives using four cancer cell lines. A couple of compounds synthesized in this study maintain the cytotoxic activity at the similar level to that of the lead compound. This reviewer thinks that the chemistry part looks good, but biology part needs more information.

The authors used four cell lines; however, they show the cytotoxic data obtained from two cell lines only in Table 1. This reviewer recommends the authors to summarize all cytotoxic data obtained from four cell lines in Table 1.

Considering the number of compounds tested on the two cell lines omitted in Table 1, the results for MDA-MB-231 have been included therein. However, results for SK-MEL-28 cell line were obtained for only four compounds, so we just decided to include the GI50 values in the text so as not to see many empty spaces in Table 1.

-The authors cultured cells in RPMI 1640 medium, which is not the optimal medium for the cell lines they used. So, this reviewer thinks that the data were obtained under a less common condition. By the way, did the authors culture cells without serum?

            Thank you for the appreciation, in fact cells were cultured in 10% of Fetal Bovine Serum (FBS). The 3.2 Biological evaluation section has been updated and the following test included: Cells are maintained in RPMI 1640 10% FBS, supplemented with 0.1 g/L penicillin and 0.1 g/L streptomycin sulfate and then incubated at 37 °C, 5% CO2 and 98% humidity.

-The authors should refer to the cytotoxic activity of the compounds on normal (non-cancer) cells because the title contains "antineoplastic." It is of importance to show that the compounds kill cancer cells specifically or nonspecifically

                The in vitro results may help to predict the usefulness of a certain molecule as a potential agent for cancer treatment. GI50's data allow to predict that not only could a drug be cytostatic, but also it could have a potential in terms of tumor reduction. Antineoplastic drugs are medications used to treat cancer and the real effects of drugs on non-cancer cells are better evaluated during and after in vivo studies on animal models. In fact, data for the anticancer napthacenequinone drug doxorubicin have also been included in Table 1 as positive control and, as can be seen, some of our compounds have GI50 values similar to those of this reference drug.

Round 2

Reviewer 1 Report

Thank you very much for the corrections made. The suggestions greatly enriched the work. For my part, I no longer require corrections.